# Seasonal influenza during the COVID-19 pandemic in Bangladesh

**Zubair Akhtar**[1]*, **Fahmida Chowdhury**[1], **Mahmudur Rahman**[1], **Probir Kumar Ghosh**[1], **Md. Kaousar Ahmmed**[1], **Md Ariful Islam**[1], **Joshua A. Mott**[2], **William Davis**[2]

**1** International Center for Diarrheal Diseases, Bangladesh, (icddr,b) Programme for Emerging Infections, Dhaka, Bangladesh, **2** Influenza Division, Centers for Disease Control and Prevention Regional Influenza Program, Bangkok, Thailand

* zakhtar@icddrb.org

**Data Availability Statement:** According to institutional data policy of the icddr,b (International Centre for Diarrhoeal Disease Research, Bangladesh), only summary of data can be publicly

## Abstract

### Introduction

During the 2019 novel coronavirus infectious disease (COVID-19) pandemic in 2020, limited data from several countries suggested reduced seasonal influenza viruses' circulation. This was due to community mitigation measures implemented to control the pandemic of severe acute respiratory syndrome coronavirus 2 (SARS-CoV-2). We used sentinel surveillance data to identify changes in the 2020 influenza season compared with previous seasons in Bangladesh.

### Methods

We used hospital-based influenza surveillance (HBIS) data of Bangladesh that are generated year-round and are population-representative severe acute respiratory infection (SARI) data for all age groups from seven public and two private tertiary care level hospitals data from 2016 to 2019. We applied the moving epidemic method (MEM) by using R language (v4.0.3), and MEM web applications (v2.14) on influenza-positive rates of SARI cases collected weekly to estimate an average seasonal influenza curve and establish epidemic thresholds.

### Results

The 2016–2019 average season started on epi week 18 (95% CI: 15–25) and lasted 12.5 weeks (95% CI: 12–14 weeks) until week 30.5. The 2020 influenza season started on epi week 36 and ended at epi week 41, lasting for only five weeks. Therefore, influenza epidemic started 18 weeks later, was 7.5 weeks shorter, and was less intense than the average epidemic of the four previous years. The 2020 influenza season started on the same week when COVID-19 control measures were halted, and 13 weeks after the measures were relaxed.

### Conclusion

Our findings suggest that seasonal influenza circulation in Bangladesh was delayed and less intense in 2020 than in previous years. Community mitigation measures may have contributed to this reduction of seasonal influenza transmission. These findings contribute to a limited but growing body of evidence that influenza seasons were altered globally in 2020.

displayed or can be made publicly accessible. To protect intellectual property rights of primary data, icddr,b cannot make primary data publicly available. However, upon request, Institutional Data Access Committee of icddr,b can provide access to primary data to any individual, upon reviewing the nature and potential use of the data. Requests for data can be forwarded to: Ms. Armana Ahmed, Head, Research Administration, icddr,b, Dhaka, Bangladesh, Email: aahmed@icddrb.org, Phone: +88 02 9827001-10 (ext. 3200).

**Funding:** This hospital-based influenza surveillance was funded by the Influenza Division of Centers for Disease Control and Prevention (CDC), Atlanta, under the co-operative agreement (6 NU51IP000852-05-01). icddr,b is grateful to the Governments of Bangladesh, Canada, Sweden, and the UK for providing core/unrestricted support. The funders had no conflict of interest with the contents of the manuscript.

**Competing interests:** The authors have declared that no competing interests exist.

## Introduction

In early 2020, the United States and several Asian countries [1–4] in the Northern Hemisphere reported sharp declines in the numbers of cases of seasonal influenza, which typically occurs during the months of October to April in these countries. In the Southern Hemisphere, Australia, Chile, South Africa and New Zealand reported similar observations during influenza season, which occurs typically from April to September in these countries [5–7]. Previous studies suggested that the decline in seasonal influenza virus activity may have been attributed to widespread community mitigation measures implemented to control the pandemic of severe acute respiratory syndrome coronavirus 2 (SARS-CoV-2) [1, 5, 7]. Further exploration of influenza seasonality in the context of SARS-CoV-2 interventions may improve our understanding of how these interventions may impact seasonal influenza transmission.

Bangladesh is a tropical country in the Northern Hemisphere, and its annual seasonal influenza epidemic occurs typically during the monsoon period, from May to September, which reflects a Southern Hemisphere pattern [8]. The SARS-CoV-2 virus emerged in Bangladesh on epidemiologic (epi) week 11 of 2020 (March 08, 2020), and pandemic control measures were initiated on March 16, 2020 (epi week 12) through closing all educational institutes and on March 26, 2020 (epi week 13) through suspending all political, religious, social, and cultural gathering including state public programs and events; closure of transport services including domestic and international flights, and closure of all public and private offices except for hospitals, kitchen markets, drug stores, and emergency services [9]. Social distancing, wearing face masks and maintaining hand hygiene were made mandatory after May 30, 2020 (from epi week 23) [10]. Enforcement of some measures was relaxed after May 30, 2020 (from epi week 23) at which time public transport service resumed (ground and air) and shopping malls/service sectors opened on a limited scale [10]. All control measures were enforced up to August 31, 2020 (epi week 36) [11].

We used sentinel surveillance data to identify changes in the 2020 influenza season compared with previous seasons in Bangladesh.

## Methods

The hospital-based influenza surveillance (HBIS) system in Bangladesh generates year-round population-representative data for all age groups from seven public and two private tertiary care level hospitals through a collaboration between the Institute of Epidemiology, Disease Control and Research (IEDCR) of the Government of Bangladesh, the International Centre for Diarrheal Diseases Research, Bangladesh (icddr,b) and the United States Centers for Disease Control and Prevention (US CDC) [12]. Participating hospitals were:

| **Public hospitals** | | |
| --- | --- | --- |
| | 1 | Rajshahi Medical College Hospital, Rajshahi |
| | 2 | Cumilla Medical College Hospital, Cumilla |
| | 3 | Khulna Medical College Hospital, Khulna |
| | 4 | Sher-e-Bangla Medical College Hospital, Barishal |
| | 5 | Chattogram Medical College Hospital, Chattogram |
| | 6 | M Abdur Rahim Medical College Hospital, Dinajpur |
| | 7 | Jashore 250 bed General Hospital, Jashore |
| **Private hospitals** | | |
| | 8 | Jahurul Islam Medical College Hospital, Kishoregonj |
| | 9 | Jalalabad Ragib-Rabeya Medical College Hospital, Sylhet |

This surveillance system has been described elsewhere, and outputs of the system include the number of Severe Acute Respiratory Infection (SARI) cases (defined as: history of fever or measured fever ≥38C with cough, onset within the previous 10 days, and requiring hospitalization) each week and the percent of those cases that tested positive for influenza viruses using real-time reverse transcription polymerase chain reaction (rRT- PCR) [12, 13]. The hospital-based influenza surveillance protocol in Bangladesh was reviewed and approved by the IRB at icddr,b. In addition, the CDC's Human Research Protection Office has reviewed and approved a continuing reliance on the icddr,b IRB.§ Participants or legal guardians provided written, informed consent before data and biological sample collection and fully anonymized data were accessed for analysis

We used the R language (v4.0.3) and MEM web applications (v2.14) to generate an average influenza seasonal curve using influenza positive rates of SARI cases collected weekly by the HBIS from January 2016 to December 2019 [14]. Briefly, this software generated epidemic curves of percent of SARI cases positive for influenza for each season of surveillance data from Bangladesh from 2016–2019. The MEM program then aligned the curves to generate an average curve, and set thresholds to define pre-epidemic, epidemic, and post-epidemic periods, as well as medium, high, and very high intensity thresholds for seasonal influenza activity [14, 15]. To determine the epidemic threshold, the MEM software calculates the upper limit of a 95% confidence interval around the 30 highest weekly values before the epidemic period, and intensity thresholds are calculated so that approximately 60% of seasons will cross the medium intensity threshold, 10% will cross the high threshold, and 2.5% will cross the very high threshold. The model estimates sensitivity (correctly defining the epidemic period above the epidemic threshold) and specificity (correctly defining the non-epidemic period below the epidemic threshold) of the model, and it calculates 95% confidence intervals for the average season's start date and duration [14]. We applied the average seasonal thresholds to identify the beginning, end, and intensity of the 2020 influenza season.

## Results

From 2016 to 2019, the HBIS reported 2,714 to 5,001 SARI cases per year; 15% to 26% of these were influenza positive. The HBIS included cases for each month of 2020, although the total of 2,361 was less than in previous years, with an average of 8% influenza positive over the entire year (Fig 1).

The MEM software generated seasonal influenza epidemic curves for 2016–2020 (Fig 2) and an average curve for 2016–2019 seasons (Fig 3). Based on the average curve, MEM identified thresholds for the start of seasonal influenza season (21% of SARI cases influenza positive), the end of the season (27% positive), and seasonal intensity (medium: 47%, high: 60%, very high: 67% positive) (Figs 1 and 2). The sensitivity of this model was 78% with 96% specificity; that is, for each season used to build the model (2016–2019), 78% of the weeks above the epidemic threshold fell within the model's predicted epidemic season and 96% of the weeks below the epidemic threshold fell within the model's predicted non-epidemic season.

The 2019 season was the most intense (peak 72% positivity), while the 2016 and 2018 seasons exceeded the high intensity threshold (peaks at 61 and 65% positivity, respectively). The 2017 season exceeded the medium intensity threshold (peak at 57% positivity), and the 2020 season exceeded the "season start" threshold but did not reach the medium intensity threshold (peak at 43% positivity, Fig 2). The 2016–2019 seasons *started on weeks 21*, *27*, *30 and 23 respectively*, and the average season started on epi week 18 (95% CI: 15–25) and lasted 12.5 weeks (95% CI: 12–14 weeks), until week 30.5 (Figs 2 and 3). The 2020 influenza season started on epi week 36 (at baseline epidemic threshold 21%) and ended at epi week 41 (at post

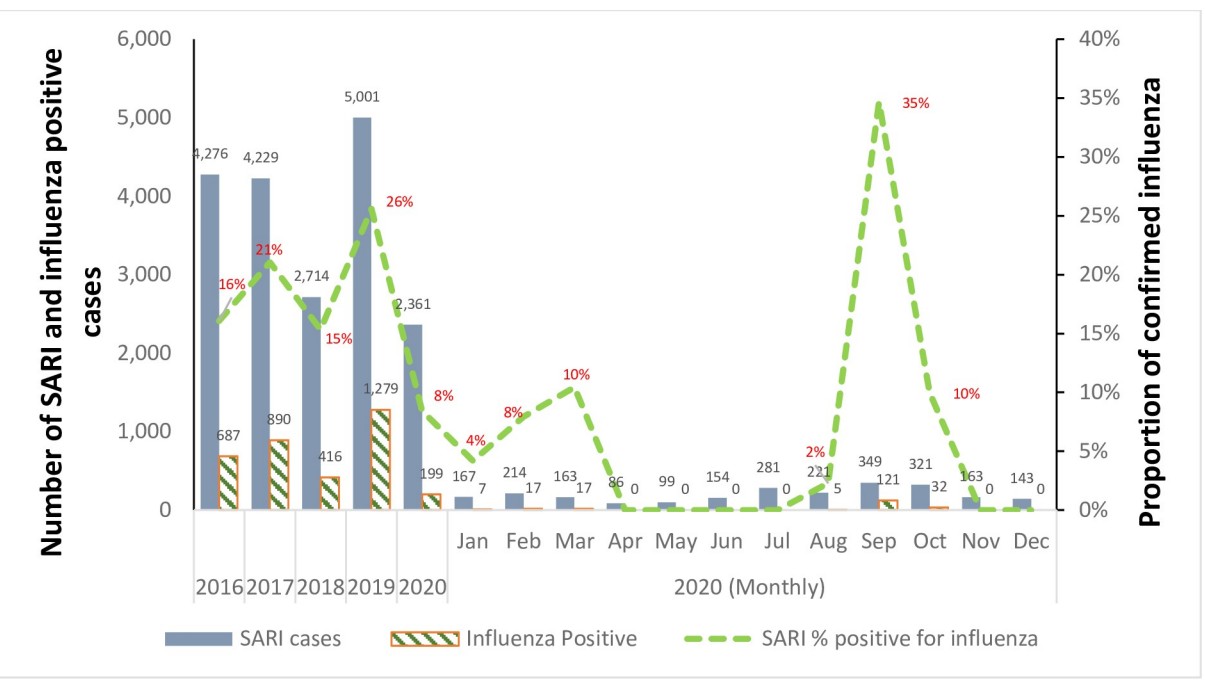

**Fig 1. SARI cases and influenza infection detected through hospital-based influenza surveillance in Bangladesh, 2016–2020.**

epidemic threshold 27%), thus lasting for only 5 weeks (Fig 4). The 2020 influenza season started on the same week COVID-19 control measures were halted, and 13 weeks after the measures were relaxed.

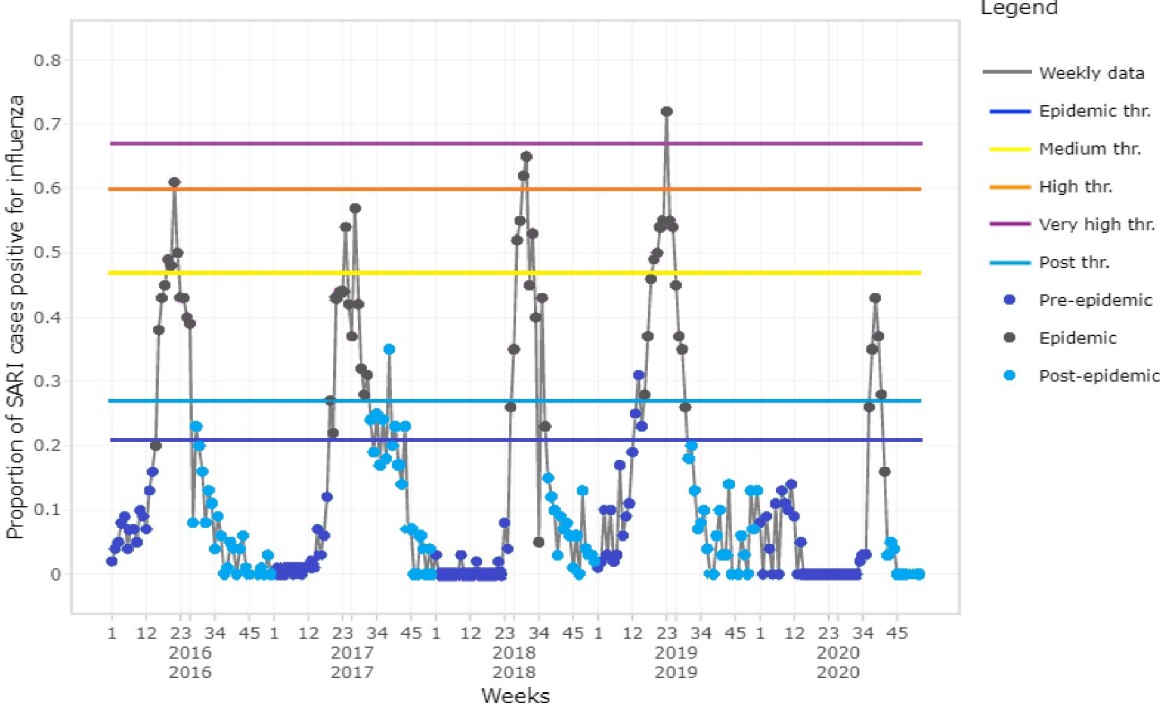

**Fig 2. Time series of influenza epidemic, the epidemic periods modelled by MEM for season 2016–2020 in Bangladesh.**

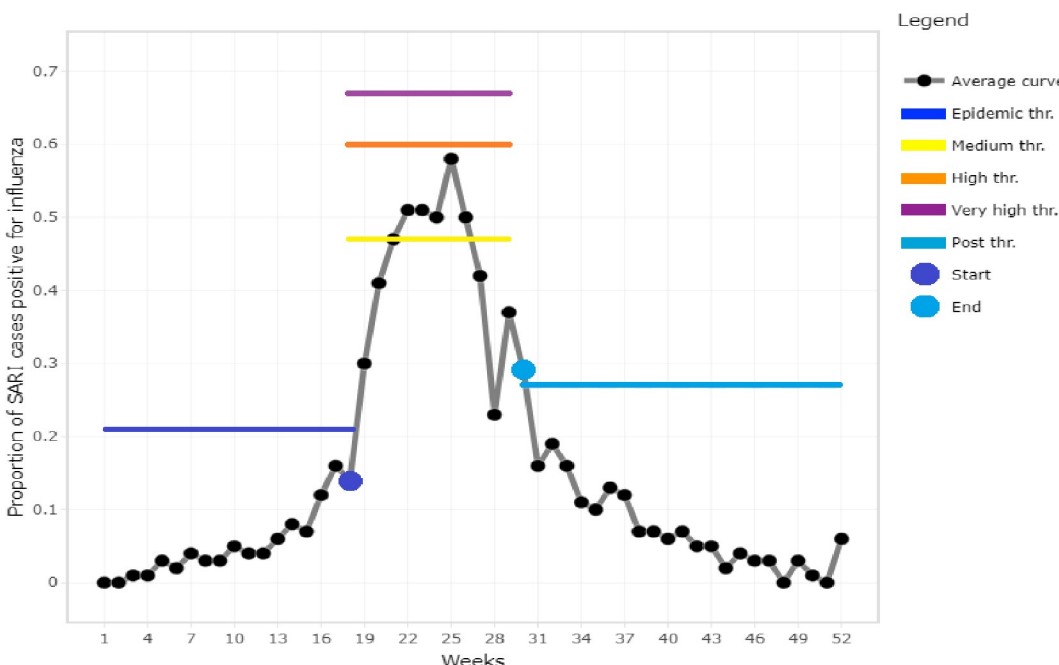

**Fig 3. Average epidemic curve and thresholds, levels of intensity and modelled influenza season 2016–2019 in Bangladesh.**

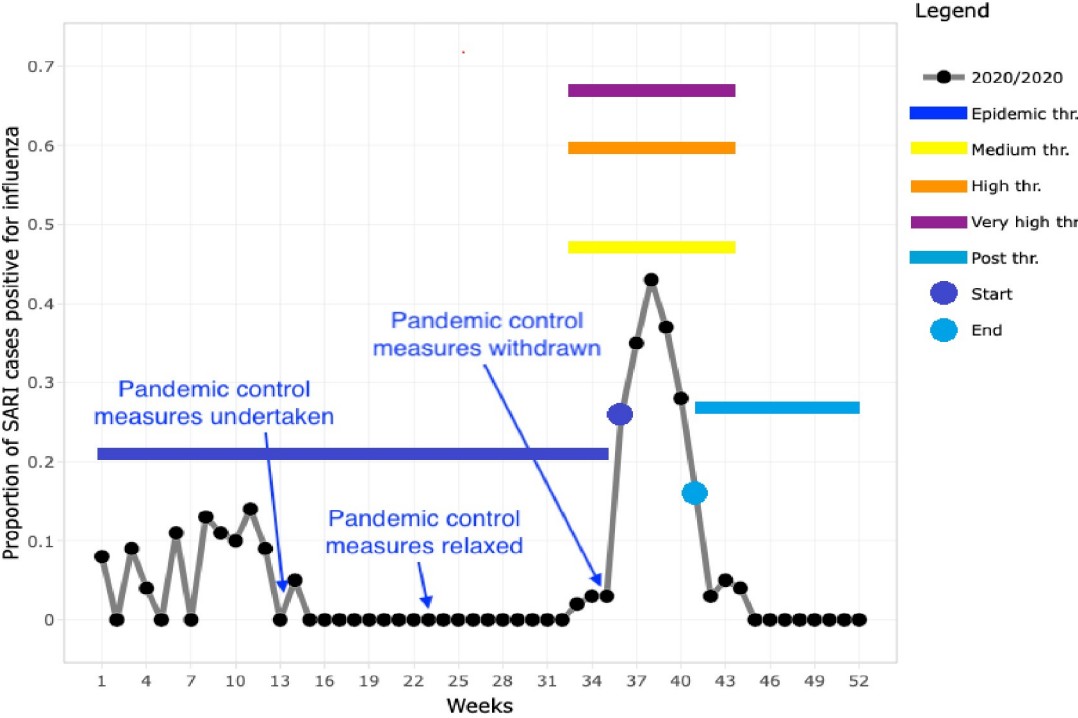

**Fig 4. Epidemic threshold, levels of intensity and modelled influenza season 2020 in Bangladesh.**

## Discussion

Bangladesh maintained influenza surveillance throughout 2020. Data indicated that the start of the 2020 influenza epidemic was 18 weeks delayed compared with the four previous seasons' average, and it lasted less than half (5 weeks vs. 12.5 weeks) the duration of previous seasons. Similar marked reductions in influenza circulation were also reported in Singapore, Thailand, China, Taiwan, Australia, South Africa, and Chile during the COVID-19 pandemic; these reports cited a collateral effect of pandemic control measures as a potential cause [1–6].

The start of the 2020 influenza season began after most of SARS-CoV-2 mitigation interventions ended. Public health control efforts undertaken to control the COVID-19 pandemic in Bangladesh were initiated five weeks before the average start of the previous four years' influenza seasons and may have contributed to delaying the country's 2020 seasonal influenza epidemic. Pandemic control measures were ultimately enforced until epi week 36 but after epi week 23 they were relaxed [11]. The 2020 influenza season started on epi week 36. Since both SARS-CoV-2 and influenza viruses have similar modes of transmission through contact, droplet, and airborne routes [16], pandemic control efforts may have also limited the transmission of influenza virus, a less transmissible virus, during its peak season in Bangladesh. Therefore, it is likely that efforts to mitigate SARS-CoV-2 contributed to the delay of the 2020 influenza season in Bangladesh.

Strengths of this report include use of data from a robust surveillance system that continued throughout the COVID-19 pandemic in 2020, and the use of an established modeling method to generate seasonal influenza epidemic timing and thresholds [7]. An important limitation is the 42% lower number of SARI cases detected by surveillance in 2020, compared with the average of the four previous seasons. This decline may have resulted from reduced health-seeking behaviors, undocumented changes in sampling of SARI patients, limited availability of personal protective equipment and reagents for sampling and testing, and logistic challenges of surveillance operations. All of these challenges may have been associated with the SARS-CoV-2 pandemic and contributed to fewer SARI patients presenting and tested for influenza and decreased the precision of relationship with influenza. Furthermore, the data presented here includes only hospital-based surveillance. Less severe influenza cases—those that did not require hospitalization—were not considered in this analysis and it is possible that trends in inpatient hospital admissions or health seeking behaviour may have changed due to the COVID-19 pandemic and confounded the influenza positivity rates. Another limitation is that we only used four seasons of data to generate the average epidemic curve; given considerable variability in influenza seasonality, more data would yield more precise start and end dates of the average season.

In conclusion, our findings suggest that seasonal influenza circulation in Bangladesh was delayed and less intense in 2020, compared with previous years. Community mitigation measures may have contributed to this reduction of seasonal influenza transmission.

## Author Contributions

**Conceptualization:** Zubair Akhtar, Mahmudur Rahman, Joshua A. Mott, William Davis.

**Data curation:** Probir Kumar Ghosh, Md. Kaousar Ahmmed, Md Ariful Islam.

**Formal analysis:** Zubair Akhtar, Probir Kumar Ghosh, Joshua A. Mott, William Davis.

**Funding acquisition:** Fahmida Chowdhury.

**Methodology:** Joshua A. Mott.

**Project administration:** Zubair Akhtar, Md. Kaousar Ahmmed, Md Ariful Islam.

**Supervision:** William Davis.

**Visualization:** Probir Kumar Ghosh.

**Writing – original draft:** Zubair Akhtar.

**Writing – review & editing:** Fahmida Chowdhury, Mahmudur Rahman, Probir Kumar Ghosh, Joshua A. Mott, William Davis.

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
