## [Decision Letter · Decision Letter 0]

25 May 2021

PONE-D-21-10066

Seasonal influenza during the COVID-19 pandemic in Bangladesh

PLOS ONE

Dear Dr. Akhtar,

Thank you for submitting your manuscript to PLOS ONE. After careful consideration, we feel that it has merit but does not fully meet PLOS ONE’s publication criteria as it currently stands. Therefore, we invite you to submit a revised version of the manuscript that addresses the points raised during the review process.

ACADEMIC EDITOR: Please see comments made by reviewers and provide point by point response in your revised manuscript

Please submit your revised manuscript by June 22nd 2021. If you will need more time than this to complete your revisions, please reply to this message or contact the journal office at plosone@plos.org. Please include the following items when submitting your revised manuscript:

We look forward to receiving your revised manuscript.

Kind regards,

Muhammad Adrish, MD, MBA, FCCP, FCCM

Academic Editor

PLOS ONE

Journal Requirements:

2)  Please include in your Methods section (or in Supplementary Information files) the participating hospitals/institutions.

3) In the Methods section and the online ethics statement, please provide clarification whether the current study reported required ethical oversight. Furthermore, please ensure that you have discussed whether all data/samples were fully anonymized before you accessed them.

Reviewers' comments:

Reviewer's Responses to Questions

**Comments to the Author**

1. Is the manuscript technically sound, and do the data support the conclusions?

Reviewer #1: Yes

Reviewer #2: Partly

Reviewer #3: Yes

2. Has the statistical analysis been performed appropriately and rigorously? 

Reviewer #1: Yes

Reviewer #2: I Don't Know

Reviewer #3: N/A

3. Have the authors made all data underlying the findings in their manuscript fully available?

Reviewer #1: No

Reviewer #2: No

Reviewer #3: Yes

4. Is the manuscript presented in an intelligible fashion and written in standard English?

Reviewer #1: Yes

Reviewer #2: Yes

Reviewer #3: Yes

5. Review Comments to the Author

Reviewer #1: This is a well written and timely report detailing reduced hospitalizations due to seasonal influenza during the COVID-19 pandemic in Bangladesh. The results are consistent with reports from other countries, and represent a valuable addition to this expanding literature. I have only one minor comment and suggestion for improvement. Since this study relied on data from a hospital based influenza surveillance system, it necessarily does not capture non-hospitalized influenza infections. Therefore, the epidemiologic picture of seasonal influenza presented is limited, and could be affected by unmeasured confounding variables, such as reduced diagnosis or hospitalization rates for severe influenza disease due to redirected public health resources. While this limitation does not invalidate this report, it should be discussed as a limitation in the Discussion section of the manuscript.

Reviewer #2: The study was considered not so satisfied in results which might be further explored and explained.

And when it comes to the question whether a statistical model could be applied to actual disease outbreak alarming, more data would be needed to do modeling.

Reviewer #3: The authors studied the changes in influenza epidemic patterns in Bangladesh during COVID-19 pandemic using hospital-based influenza surveillance data of Bangladesh from 2016-2020 and reported that 2020 influenza season was delayed and less intense than the previous years. I have some questions about data analysis.

How is the average curve calculated? Was a function in the R mem package used to align the curves of seasons 2016-2019 to generate the average curve? Why were the average seasonal thresholds of 2016-2019 used to identify the beginning, ending and intensity of 2020 influenza season instead of letting the data from 2020 determine the thresholds.

Please give the MEM results (refer to tables in references 14, 15) for each season separately.

Why is the last black point (2020 around weeks 40-45, y-axis is round 0.18) in Figure 2 black (epidemic) instead of blue (post-epidemic) since it is already lower than the post-threshold. Also shouldn’t the start of the epidemic be the first week where the rate is higher than pre-threshold? The black points or blue point (Figure 3) seem to start before the rate reaches pre-epidemic threshold.

Last sentence on pages 10-11: “MEM software defines the beginning and end of the epidemic periods as the times at which the rate of positive cases is higher than optimized seasonal data, ..” shouldn’t “optimizes seasonal data” be “pre-epidemic threshold”?

6. PLOS authors have the option to publish the peer review history of their article (what does this mean?). If published, this will include your full peer review and any attached files.

Reviewer #1: No

Reviewer #2: No

Reviewer #3: No

---

## [Author Response · Author response to Decision Letter 0]

6 Jun 2021

Reviewer 1 comment

This is a well written and timely report detailing reduced hospitalizations due to seasonal influenza during the COVID-19 pandemic in Bangladesh. The results are consistent with reports from other countries, and represent a valuable addition to this expanding literature. I have only one minor comment and suggestion for improvement. Since this study relied on data from a hospital based influenza surveillance system, it necessarily does not capture non-hospitalized influenza infections. Therefore, the epidemiologic picture of seasonal influenza presented is limited, and could be affected by unmeasured confounding variables, such as reduced diagnosis or hospitalization rates for severe influenza disease due to redirected public health resources. While this limitation does not invalidate this report, it should be discussed as a limitation in the Discussion section of the manuscript.

Response: Thank you for your appreciating and thoughtful comment about a significant limitation. We agree that the less severe cases not requiring hospitalization and redirected health resources may have influenced our results but did not invalidate our findings. Thus we have added another limitation addressing this in our discussion section as following:

“Furthermore, the data presented here includes only hospital-based surveillance. Less severe influenza cases-- those that did not require hospitalization-- were not considered in this analysis and it is possible that trends in inpatient hospital admissions or health seeking behaviour may have changed due to the COVID-19 pandemic and confounded the influenza positivity rates.”

Reviewer 2 comment

The study was considered not so satisfied in results which might be further explored and explained. And when it comes to the question whether a statistical model could be applied to actual disease outbreak alarming, more data would be needed to do modeling

Response: We thank the reviewer for this thoughtful critique of the statistical model. Since we are describing the influenza season of 2020 only, we restricted our analysis to the previous four years because the MEM model achieved a sensitivity of 78% with 96% specificity. These sensitivity and specificity were above the limits of a robust model as cross-validated in previous literature (reference #14 in manuscript, Vega, 2013). We agree with the reviewer that when this model would be applied for actual forecasting, more data would yield a more precise start of an epidemic. If we had used this model to predict future influenza outbreaks in a paper, e.g., that of 2021, we would have certainly opted to use more data.

 

Reviewer 3 comments

The authors studied the changes in influenza epidemic patterns in Bangladesh during COVID-19 pandemic using hospital-based influenza surveillance data of Bangladesh from 2016-2020 and reported that 2020 influenza season was delayed and less intense than the previous years. I have some questions about data analysis.

1. How is the average curve calculated? Was a function in the R mem package used to align the curves of seasons 2016-2019 to generate the average curve? Why were the average seasonal thresholds of 2016-2019 used to identify the beginning, ending and intensity of 2020 influenza season instead of letting the data from 2020 determine the thresholds.

Response: Thank you for seeking clarification. The function in the R MEM package generated the generated seasonal influenza epidemic curves for 2016-2020 (Figure 2). Following this, epidemic thresholds are calculated, and average thresholds based on the 2016-2019 seasons are illustrated in Figure 3. Since we are describing season 2020 relative to previous years (2016-2019), in figure 4, actual 2020 data is used to compare with threshold values set by 2016-2019 data.

2. Please give the MEM results (refer to tables in references 14, 15) for each season separately.

Response: We appreciate your interest in moving epidemic method (MEM). The tables added in reference 14 and 15 describe the MEM and have been published establishing the MEM in describing influenza seasons. Figures 2 and 3 show individual seasons and the average curve generated from these seasons’ data. We added information to the text describing start weeks and peak intensities: 

“The 2019 season was the most intense (peak 72% positivity), while the 2016 and 2018 seasons exceeded the high intensity threshold (peaks at 61 and 65% positivity, respectively). The 2017 season exceeded the medium intensity threshold (peak at 57%), and the 2020 season exceeded the “season start” threshold but did not reach the medium intensity threshold (peak at 43% positivity, Fig 2). The 2016-2019 seasons started on weeks 21, 27, 30 and 23 respectively, and the average season started on epi week 18 (95% CI: 15-25) and lasted 12.5 weeks (95% CI: 12-14 weeks), until week 30.5 (Figs 2 and 3).”

3. Why is the last black point (2020 around weeks 40-45, y-axis is round 0.18) in Figure 2 black (epidemic) instead of blue (post-epidemic) since it is already lower than the post-threshold. Also shouldn't the start of the epidemic be the first week where the rate is higher than pre-threshold? The black points or blue point (Figure 3) seem to start before the rate reaches pre-epidemic threshold.

Response: We appreciate your discreet scrutiny of the figures. The curves generated in Fig 2 are based on actual data points (proportion positives) of the respective years. Here deep blue ones are pre-epidemic, black/grey ones are epidemic, and light blue/cyan is post-epidemic data points. The threshold lines generated are based on the average of the 2016-2019 seasons, and data in the 2020 season is smoothed for analysis. Because data are smoothed for analyses when points fall very close to the thresholds, the smoothing may place them in a different threshold category from where they appear to be on the graph. 

4. Last sentence on pages 10-11: "MEM software defines the beginning and end of the epidemic periods as the times at which the rate of positive cases is higher than optimized seasonal data, .." shouldn't "optimizes seasonal data" be "pre-epidemic threshold"?

Response: We added some more detail to the methods, to include this, “To estimate the epidemic threshold, the MEM software calculates the upper limit of a 95% confidence interval around the 30 highest weekly values before the epidemic period,” we also added a reference to the Vega 2013 paper which describes the MEM methodology in detail.

---

## [Decision Letter · Decision Letter 1]

21 Jul 2021

Seasonal influenza during the COVID-19 pandemic in Bangladesh

PONE-D-21-10066R1

Dear Dr. Akhtar,

We’re pleased to inform you that your manuscript has been judged scientifically suitable for publication and will be formally accepted for publication once it meets all outstanding technical requirements.

Kind regards,

Muhammad Adrish, MD, MBA, FCCP, FCCM

Academic Editor

PLOS ONE

Additional Editor Comments (optional):

All comments have been addressed.

Reviewers' comments:

Reviewer's Responses to Questions

**Comments to the Author**

1. If the authors have adequately addressed your comments raised in a previous round of review and you feel that this manuscript is now acceptable for publication, you may indicate that here to bypass the “Comments to the Author” section, enter your conflict of interest statement in the “Confidential to Editor” section, and submit your "Accept" recommendation.

Reviewer #1: All comments have been addressed

Reviewer #3: All comments have been addressed

2. Is the manuscript technically sound, and do the data support the conclusions?

Reviewer #1: (No Response)

Reviewer #3: Yes

3. Has the statistical analysis been performed appropriately and rigorously? 

Reviewer #1: (No Response)

Reviewer #3: Yes

4. Have the authors made all data underlying the findings in their manuscript fully available?

Reviewer #1: (No Response)

Reviewer #3: No

5. Is the manuscript presented in an intelligible fashion and written in standard English?

Reviewer #1: (No Response)

Reviewer #3: Yes

6. Review Comments to the Author

Reviewer #1: (No Response)

Reviewer #3: (No Response)

7. PLOS authors have the option to publish the peer review history of their article (what does this mean?). If published, this will include your full peer review and any attached files.

Reviewer #1: No

Reviewer #3: No

---

## [Editor Report · Acceptance letter]

26 Jul 2021

PONE-D-21-10066R1 

Seasonal influenza during the COVID-19 pandemic in Bangladesh 

Dear Dr. Akhtar:

I'm pleased to inform you that your manuscript has been deemed suitable for publication in PLOS ONE. Congratulations! Your manuscript is now with our production department. 

Kind regards, 

on behalf of

Dr. Muhammad Adrish 

Academic Editor

PLOS ONE